# Rethinking Positional Encoding in Language Pre-training

**Guolin Ke, Di He & Tie-Yan Liu**
Microsoft Research*
{guolin.ke, dihe, tyliu}@microsoft.com

## Abstract

In this work, we investigate the positional encoding methods used in language pre-training (e.g., BERT) and identify several problems in the existing formulations. First, we show that in the absolute positional encoding, the addition operation applied on positional embeddings and word embeddings brings mixed correlations between the two heterogeneous information resources. It may bring unnecessary randomness in the attention and further limit the expressiveness of the model. Second, we question whether treating the position of the symbol [CLS] the same as other words is a reasonable design, considering its special role (the representation of the entire sentence) in the downstream tasks. Motivated from above analysis, we propose a new positional encoding method called **T**ransformer with **U**ntied **P**ositional **E**ncoding (TUPE). In the self-attention module, TUPE computes the word contextual correlation and positional correlation separately with different parameterizations and then adds them together. This design removes the mixed and noisy correlations over heterogeneous embeddings and offers more expressiveness by using different projection matrices. Furthermore, TUPE unties the [CLS] symbol from other positions, making it easier to capture information from all positions. Extensive experiments and ablation studies on GLUE benchmark demonstrate the effectiveness of the proposed method. Codes and models are released at https://github.com/guolinke/TUPE.

## 1 Introduction

The Transformer model (Vaswani et al., 2017) is the most widely used architecture in language representation learning (Liu et al., 2019; Devlin et al., 2018; Radford et al., 2019; Bao et al., 2020). In Transformer, positional encoding is an essential component since other main components of the model are entirely invariant to sequence order. The original Transformer uses the absolute positional encoding, which provides each position an embedding vector. The positional embedding is added to the word embedding, which is found significantly helpful at learning the contextual representations of words at different positions. Besides using the absolute positional encoding, Shaw et al. (2018); Raffel et al. (2019) further propose the relative positional encoding, which incorporates some carefully designed bias term inside the self-attention module to encode the distance between any two positions.

In this work, we revisit and study the formulation of the widely used absolute/relative positional encoding. First, we question the rationality of adding the word embedding with the absolute positional embedding in the input layer. Since the two kinds of embeddings are apparently heterogeneous, this addition operation brings mixed correlations[1] between the positional information and word semantics. For example, by expanding the dot-production function of keys and values in the self-attention module of the first layer, we find that there are explicit terms that use "word" to query "positions" and vice versa. However, words may only have weak correlations to where they appear in the sentence. Our empirical analysis also supports this by showing that in a well-trained model, such correlation is noisy.

Second, we notice that the BERT model does not only handle natural language words. A special symbol [CLS] is usually attached to the sentence. It is widely acknowledged that this symbol

---

*Correspondence to:{guolin.ke, dihe}@microsoft.com
[1]The term "correlation" mainly refers to the dot product between Key and Query in the self-attention module.

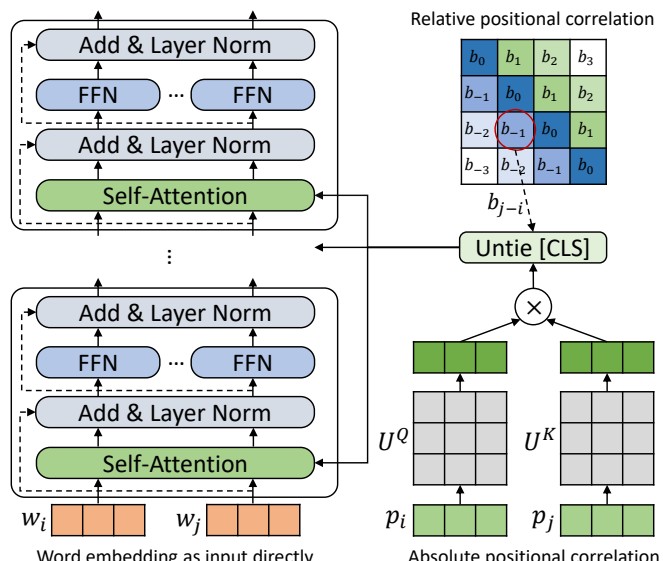

Figure 1: The architecture of TUPE. The positional correlation and word correlation are computed separately, and added together in the self-attention module. The positional attention related to the [CLS] token is treated more positionless, to encourage it captures the global information.

receives and summarizes useful information from all the positions, and the contextual representation of [CLS] will be used as the representation of the sentence in the downstream tasks. As the role of the [CLS] symbol is different from regular words that naturally contain semantics, we argue that it will be ineffective if we treat its position the same as word positions in the sentence. For example, if we apply the relative positional encoding to this symbol, the attention distribution of some heads will likely be biased to the first several words, which hurts the understanding of the whole sentence.

Based on the investigation above, we propose several simple, yet effective modifications to the current methods, which lead to a new positional encoding called **T**ransformer with **U**ntied **P**ositional **E**ncoding (TUPE) for language pre-training, see Figure 1. In TUPE, the Transformer only uses the word embedding as input. In the self-attention module, different types of correlations are separately computed to reflect different aspects of information, including word contextual correlation and absolute (and relative) positional correlation. Each kind of correlation has its own parameters and will be added together to generate the attention distribution. A specialized positional correlation is further set to the [CLS] symbol, aiming to capture the global representation of the sentence correctly. First, we can see that in TUPE, the positional correlation and word contextual correlation are de-coupled and computed using different parameters. This design successfully removes the randomness in word-to-position (or position-to-word) correlations and gives more expressiveness to characterize the relationship between a pair of words or positions. Second, TUPE uses a different function to compute the correlations between the [CLS] symbol and other positions. This flexibility can help the model learn an accurate representation of the whole sentence.

We provide an efficient implementation of TUPE. To validate the method, we conduct extensive experiments and ablation studies on the GLUE benchmark dataset. Empirical results confirm that our proposed TUPE consistently improves the model performance on almost all tasks. In particular, we observe that by imposing this inductive bias to encode the positional information, the model can be trained more effectively, and the training time of the pre-training stage can be largely improved.

## 2 PRELIMINARY

### 2.1 ATTENTION MODULE

The attention module (Vaswani et al., 2017) is formulated as querying a dictionary with key-value pairs, e.g., $\text{Attention}(Q, K, V) = \text{softmax}(\frac{QK^T}{\sqrt{d}})V$, where $d$ is the dimensionality of the hidden

representations, and $Q$ (Query), $K$ (Key), $V$ (Value) are specified as the hidden representations of the previous layer. The multi-head variant of the attention module is popularly used which allows the model to jointly attend to the information from different representation sub-spaces, and is defined as

$$\text{Multi-head}(Q, K, V) = \text{Concat}(\text{head}_1, \cdots, \text{head}_H)W^O$$
$$\text{head}_k = \text{Attention}(QW_k^Q, KW_k^K, VW_k^V), \tag{1}$$

where $W_k^Q \in \mathbb{R}^{d \times d_K}, W_k^K \in \mathbb{R}^{d \times d_K}, W_k^V \in \mathbb{R}^{d \times d_V}$, and $W^O \in \mathbb{R}^{Hd_V \times d}$ are learnable project matrices, $H$ is the number of heads. $d_K$ and $d_V$ are the dimensionalities of Key and Value.

The self-attention module is one of the key components in Transformer and BERT encoder (Devlin et al., 2018). For simplicity, we use the single-head self-attention module and set $d_K = d_V = d$ for a demonstration. We denote $x^l = (x_1^l, x_2^l \cdots, x_n^l)$ as the input to the self-attention module in the $l$-th layer, where $n$ is the length of the sequence and each vector $x_i^l \in \mathbb{R}^d$ is the contextual representation of the token at position $i$. $z^l = (z_1^l, z_2^l \cdots, z_n^l)$ is the output of the attention module. Then the self-attention module can be written as

$$z_i^l = \sum_{j=1}^n \frac{\exp(\alpha_{ij})}{\sum_{j'=1}^n \exp(\alpha_{ij'})}(x_j^l W^{V,l}), \text{ where } \alpha_{ij} = \frac{1}{\sqrt{d}}(x_i^l W^{Q,l})(x_j^l W^{K,l})^T. \tag{2}$$

As we can see, the self-attention module does not make use of the order of the sequence, i.e., is permutation-invariant. However, natural language is well-structured and word order is important for language understanding (Sutskever et al., 2014). In the next section, we show several previous works that proposed different ways of incorporating positional information into the Transformer model.

## 2.2 POSITIONAL ENCODING

Generally, there are two categories of methods that encode positional information in the Transformer model, absolute positional encoding and relative positional encoding.

**Absolute Positional Encoding.** The original Transformer (Vaswani et al., 2017) proposes to use absolute positional encoding to represent positions. In particular, a (learnable) real-valued vector $p_i \in \mathbb{R}^d$ is assigned to each position $i$. Given a sentence, $p_i$ will be added to the word embedding $w_i$ at position $i$, and $w_i + p_i$ will be used as the input to the model, e.g, $x_i^1 = w_i + p_i$. In such a way, the Transformer can differentiate the words coming from different positions and assign each token position-dependent attention. For example, in the self-attention module in the first layer, we have

$$\alpha_{ij}^{Abs} = \frac{1}{\sqrt{d}}((w_i + p_i)W^{Q,1})((w_j + p_j)W^{K,1})^T. \tag{3}$$

**Relative Positional Encoding.** In absolute positional encoding, using different $p_i$ for different position $i$ helps Transformer distinguish words at different positions. However, as pointed out in Shaw et al. (2018), the absolute positional encoding is not effective for the model to capture the relative word orders. Therefore, besides using absolute positional encoding, Shaw et al. proposes a relative positional encoding as an inductive bias to help the learning of attention modules,

$$\alpha_{ij}^{Rel} = \frac{1}{\sqrt{d}}(x_i^l W^{Q,l})(x_j^l W^{K,l} + a_{j-i}^l)^T, \tag{4}$$

where $a_{j-i}^l \in \mathbb{R}^d$ is learnable parameter and can be viewed as the embedding of the relative position $j - i$ in layer $l$. In this way, embedding $a_{j-i}^l$ explicitly models the relative word orders. T5 (Raffel et al., 2019) further simplifies it by eliminating $a_{j-1}^l$ in Query-Key product.

$$\alpha_{ij}^{T5} = \frac{1}{\sqrt{d}}(x_i^l W^{Q,l})(x_j^l W^{K,l})^T + b_{j-i}. \tag{5}$$

For each $j - i$, $b_{j-i}$ is a learnable scalar[2] and shared in all layers.

---

[2]Specifically, in Shaw et al. (2018); Raffel et al. (2019), the relative position $j - i$ will be first clipped to a pre-defined range, e.g., $\text{clip}(j - i, -t, t)$, $t = 128$. The embedding is defined over the possible values of the clipped range, i.e., $[-t, t]$. Besides, Shaw et al. also tried to add vector $a_{j-i}^{V,l}$ to the value $V$ in the output of self-attention, but the experiment results indicate that it did not improve much.

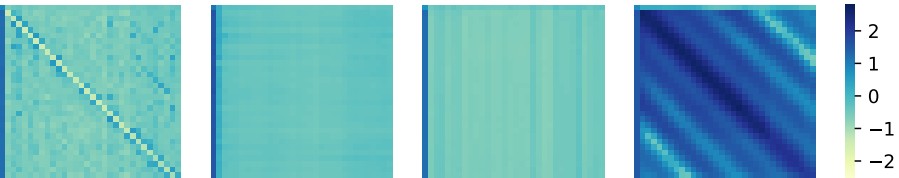

Figure 2: Visualizations of the four correlations (Eq. (6)) on a pre-trained BERT model for a sampled batch of sentences. From left to right: word-to-word, word-to-position, position-to-word, and position-to-position correlation matrices. In each matrix, the ($i$-th, $j$-th) element is the correlation between $i$-th word/position and $j$-th word/position. We can find that the correlations between a word and a position are not strong since the values in the second and third matrices look uniform.

## 3 TRANSFORMER WITH UNTIED POSITIONAL ENCODING

### 3.1 UNTIE THE CORRELATIONS BETWEEN POSITIONS AND WORDS

In absolute positional encoding, the positional embedding is added together with the word embedding to serves as the input to the neural networks. However, these two kinds of information are heterogeneous. The word embedding encodes the semantic meanings of words and word analogy tasks can be solved using simple linear arithmetic on word embeddings (Mikolov et al., 2013; Pennington et al., 2014; Joulin et al., 2016). On the other hand, the absolute positional embedding encodes the indices in a sequence, which is not semantic and far different from word meanings. We question the rationality of the linear arithmetic between the word embedding and the positional embedding. To check clearly, we take a look at the expansion of Eq. (3).

$$
\begin{aligned}
\alpha_{ij}^{Abs} &= \frac{((w_i + p_i)W^{Q,1})((w_j + p_j)W^{K,1})^T}{\sqrt{d}} \\
&= \frac{(w_i W^{Q,1})(w_j W^{K,1})^T}{\sqrt{d}} + \frac{(w_i W^{Q,1})(p_j W^{K,1})^T}{\sqrt{d}} \\
&+ \frac{(p_i W^{Q,1})(w_j W^{K,1})^T}{\sqrt{d}} + \frac{(p_i W^{Q,1})(p_j W^{K,1})^T}{\sqrt{d}}
\end{aligned}
\tag{6}
$$

The above expansion shows how the word embedding and the positional embedding are projected and queried in the attention module. We can see that there are four terms after the expansion: word-to-word, word-to-position, position-to-word, and position-to-position correlations.

We have several concerns regarding this formulation. First, it is easy to see that the first and the last term characterize the word-word and position-position relationships respectively. However, the projection matrices $W^{Q,l}$ and $W^{K,l}$ are shared in both terms. As the positional embedding and the word embedding encode significantly different concepts, it is not reasonable to apply the same projection to such different information.

Furthermore, we also notice that the second and the third term use the position (word) as the query to get keys composed of words (positions). As far as we know, there is little evidence suggesting that the word and its location in a sentence have a strong correlation. Furthermore, in BERT and recently developed advanced methods such as RoBERTa (Liu et al., 2010), sentences are patched in a random way. For example, in BERT, each input contains multiple sentences and some of the sentences are negatively sampled from other documents to form the next sentence prediction task. Due to the random process of batching, it is possible that a word can even appear at any positions and the correlations between words and positions could be weak. To further investigate this, we visualize the four correlations in Eq. (6) on a pre-trained BERT model. We find that the second and the third term looks uniform across positions, as shown in Figure 2. This phenomenon suggests that there are no strong correlations[3] between the word and the absolute position and using such noisy correlation may be inefficient for model training.

---

[3]Some recent works (Yang et al., 2019; He et al., 2020) show that correlations between relative positions and words can improve the performance. Our results have no contradiction with theirs as our study is on the correlations between word embeddings and absolute positional embeddings.

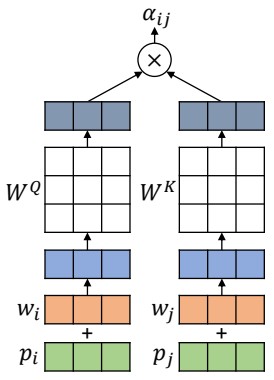
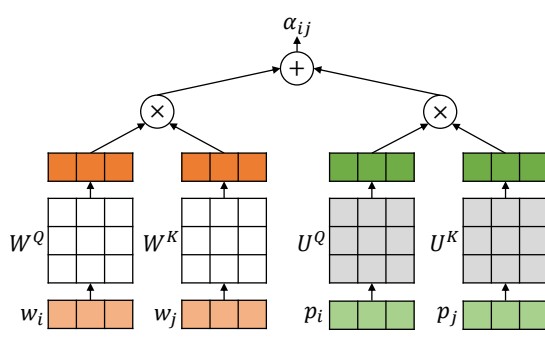

(a) Absolute positional encoding.
(b) Untied absolute positional encoding.

Figure 3: Instead of adding the absolute positional embedding to the word embedding in the input (left), we compute the positional correlation and word correlation separately with different projection matrices, and add them together in the self-attention module (right).

**Our modification.** To overcome these problems, we propose to directly model the relationships between a pair of words or positions by using different projection matrices and remove the two terms in the middle. That is, we use

$$\alpha_{ij} = \frac{1}{\sqrt{2d}}(x_i^l W^{Q,l})(x_j^l W^{K,l})^T + \frac{1}{\sqrt{2d}}(p_i U^Q)(p_j U^K)^T, \tag{7}$$

where $U^Q, U^K \in \mathbb{R}^{d \times d}$ are the projection matrice for the positional embedding, and scaling term $\frac{1}{\sqrt{2d}}$ is used to retain the magnitude of $\alpha_{ij}$ (Vaswani et al., 2017) . A visualization is put in Figure 3. Our proposed method can be well combined with the relative positional encoding in Raffel et al. (2019) by simply changing Eq. (5) to

$$\alpha_{ij} = \frac{1}{\sqrt{2d}}(x_i^l W^{Q,l})(x_j^l W^{K,l})^T + \frac{1}{\sqrt{2d}}(p_i U^Q)(p_j U^K)^T + b_{j-i}. \tag{8}$$

### 3.2 UNTIE THE [CLS] SYMBOL FROM POSITIONS

Note that in language representation learning, the input sequence to the Transformer model is not always a natural sentence. In BERT, a special symbol [CLS] is attached to the beginning of the input sentence. This symbol is designed to capture the *global* information of the whole sentence. Its contextual representation will be used to make predictions in the sentence-level downstream tasks after pre-training (Devlin et al., 2018; Liu et al., 2019).

We argue that there will be some disadvantages if we treat this token the same as other natural words in the attention module. For example, the regular words usually have strong *local* dependencies in the sentence. Many visualizations (Clark et al., 2019a; Gong et al., 2019) show that the attention distributions of some heads concentrate locally. If we process the position of [CLS] the same as the position of natural language words, according to the aforementioned local concentration phenomenon, [CLS] will be likely biased to focus on the first several words instead of the whole sentence. It will potentially hurt the performance of the downstream tasks.

**Our modification.** We give a specific design in the attention module to untie the [CLS] symbol from other positions. In particular, we reset the positional correlations related to [CLS]. For better demonstration, we denote $v_{ij}$ as the content-free (position-only) correlation between position $i$ and $j$. For example, when using the absolute positional encoding in Eq. (7), $v_{ij} = \frac{1}{\sqrt{2d}}(p_i U^Q)(p_j U^K)^T$; when using relative positional encoding in Eq. (8), $v_{ij} = \frac{1}{\sqrt{2d}}(p_i U^Q)(p_j U^K)^T + b_{j-i}$. We reset the values of $v_{ij}$ by the following equation:

$$\text{reset}_\theta(v, i, j) = \begin{cases} v_{ij} & i \neq 1, j \neq 1, (\text{not related to [CLS]}) \\ \theta_1 & i = 1, (\text{from [CLS] to others}) \\ \theta_2 & i \neq 1, j = 1, (\text{from others to [CLS]}) \end{cases}, \tag{9}$$

where $\theta = \{\theta_1, \theta_2\}$ is a learnable parameter. A visualization is put in Figure 4.

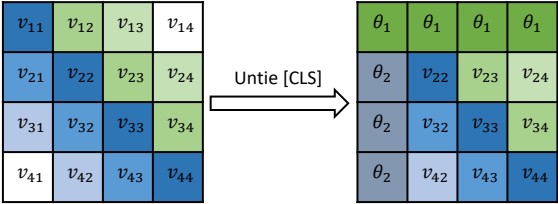

Figure 4: Illustration of untying [CLS]. $v_{ij}$ denotes the positional correlation of pair $(i, j)$. The first row and first column are set to the same values respectively.

### 3.3 IMPLEMENTATION DETAILS AND DISCUSSIONS

In the two subsections above, we propose several modifications to untie the correlations between positions and words (Eq. (7) and Eq. (8)), and untie the [CLS] symbol from other positions (Eq. (9)). By combining them, we obtain a new positional encoding method and call it TUPE (Transformer with Untied Positional Encoding). There are two versions of TUPE. The first version is to use the untied absolute positional encoding with the untied [CLS] symbol (Eq. (7) + Eq. (9)), and the second version is to use an additional relative positional encoding (Eq. (8) + Eq. (9)). We call them TUPE-A and TUPE-R respectively and list the mathematical formulations as below.

$$\alpha_{ij}^{\text{TUPE-A}} = \frac{1}{\sqrt{2d}}(x_i^l W^{Q,l})(x_j^l W^{K,l})^T + \text{reset}_\theta(\frac{1}{\sqrt{2d}}(p_i U^Q)(p_j U^K)^T, i, j) \tag{10}$$

$$\alpha_{ij}^{\text{TUPE-R}} = \frac{1}{\sqrt{2d}}(x_i^l W^{Q,l})(x_j^l W^{K,l})^T + \text{reset}_\theta(\frac{1}{\sqrt{2d}}(p_i U^Q)(p_j U^K)^T + b_{j-i}, i, j), \tag{11}$$

**The multi-head version, parameter sharing, and efficiency.** TUPE can be easily extended to the multi-head version. In our implementation, the absolute positional embedding $p_i$ for position $i$ is shared across different heads, while for each head, the projection matrices $U^Q$ and $U^K$ are different. For the relative positional encoding, $b_{j-i}$ is different for different heads. The reset parameter $\theta$ is also not shared across heads. For efficiency, we share the (multi-head) projection matrices $U^Q$ and $U^K$ in different layers. Therefore, in TUPE, the number of total parameters does not change much. Taking BERT-Base as an example, we introduce about 1.18M $(2 \times 768 \times 768)$ new parameters, which is only about 1% of the 110M parameters in BERT-Base. Besides, TUPE introduces little additional computational costs. As the positional correlation term $\frac{1}{\sqrt{2d}}(p_i U^Q)(p_j U^K)^T$ is shared in all layers, we only need to compute it in the first layer, and reuse its outputs in other layers.

**Are absolute/relative positional encoding redundant to each other?** One may think that both the absolute/relative positional encoding in Eq. (11) describe the content-free correlation, and thus one of them is redundant. To formally study this, we denote $B$ as an $n \times n$ matrix where each element $B_{i,j} = b_{j-i}$. By definition, $B$ is a Toeplitz matrix (Gray, 2006). We also denote $P$ as an $n \times n$ matrix where the $i$-th row is $p_i$, and thus the absolute positional correlation in matrix form is $\frac{1}{\sqrt{2d}}(PU^Q)(PU^K)^T$. We study the expressiveness of $B$ and $\frac{1}{\sqrt{2d}}(PU^Q)(PU^K)^T$ by first showing $B$ can be factorized similarly from the following proposition.

**Proposition 1.** *Any Toeplitz matrix $B \in C^{n \times n}$ can be factorized into $B = GDG^*$, where $D$ is a $2n \times 2n$ diagonal matrix. $G$ is a $n \times 2n$ Vandermonde matrix in the complex space, where each element $G_{j,k} = \frac{1}{2n}e^{i\pi(j+1)k/n}$ and $G^*$ is the conjugate transpose of $G$.*

The proof can be found in Appendix A. The two terms $B$ and $\frac{1}{\sqrt{2d}}(PU^Q)(PU^K)^T$ actually form different subspaces in $R^{n \times n}$. In the multi-head version, the shape of the matrix $U^Q$ and $U^k$ are $d \times \frac{d}{H}$. Therefore, $(PU^Q)(PU^K)^T$ can characterize low-rank matrices in $R^{n \times n}$. But from the proposition, we can see that $B$ forms a linear subspace in $R^{n \times n}$ with only $2n - 1$ freedoms, which is quite different from the space of $\frac{1}{\sqrt{2d}}(PU^Q)(PU^K)^T$. There are also some practical reasons which make using both terms together essential. As discussed previously, in Raffel et al. (2019), the range of the relative distance $j - i$ will be clipped up to an offset beyond which all relative positions will be assigned the same value. In such a situation, the relative positional encoding may not be able to differentiate words faraway and $\frac{1}{\sqrt{2d}}(p_i U^Q)(p_j U^K)^T$ can be used to encode complementary information.

Table 1: GLUE scores on dev set. All settings are pre-trained by BERT-Base (110M) model with 16GB data. TUPE-A$^{\text{mid}}$ (TUPE-R$^{\text{mid}}$) is the intermediate $300k$-step checkpoint of TUPE-A (TUPE-R). TUPE-A$^{\text{tie-cls}}$ removes the reset function from TUPE-A. BERT-A$^d$ uses different projection matrices for words and positions, based on BERT-A.

| | Steps | MNLI-m/mm | QNLI | QQP | SST | CoLA | MRPC | RTE | STS | Avg. |
|---|---|---|---|---|---|---|---|---|---|---|
| BERT-A | $1M$ | 84.93/84.91 | 91.34 | 91.04 | 92.88 | 55.19 | 88.29 | 68.61 | **89.43** | 82.96 |
| BERT-R | $1M$ | 85.81/85.84 | 92.16 | 91.12 | 92.90 | 55.43 | 89.26 | 71.46 | 88.94 | 83.66 |
| TUPE-A | $1M$ | 86.05/85.99 | 91.92 | 91.16 | 93.19 | 63.09 | 88.37 | 71.61 | 88.88 | 84.47 |
| TUPE-R | $1M$ | **86.21/86.19** | **92.17** | **91.30** | **93.26** | **63.56** | **89.89** | **73.56** | 89.23 | **85.04** |
| TUPE-A$^{\text{mid}}$ | $300k$ | 84.76/84.83 | 90.96 | 91.00 | 92.25 | 62.13 | 87.1 | 68.79 | 88.16 | 83.33 |
| TUPE-R$^{\text{mid}}$ | $300k$ | 84.86/85.21 | 91.23 | 91.14 | 92.41 | 62.47 | 87.29 | 69.85 | 88.63 | 83.68 |
| TUPE-A$^{\text{tie-cls}}$ | $1M$ | 85.91/85.73 | 91.90 | 91.05 | 93.17 | 59.46 | 88.53 | 69.54 | 88.97 | 83.81 |
| BERT-A$^d$ | $1M$ | 85.26/85.28 | 91.56 | 91.02 | 92.70 | 59.73 | 88.46 | 71.31 | 87.47 | 83.64 |

# 4 EXPERIMENT

To verify the performance of the proposed TUPE, we conduct extensive experiments and demonstrate the results in this section. In the main body of the paper, we study TUPE under the BERT-Base setting (Devlin et al., 2018). We provide all the experimental details and more results on applying TUPE under the BERT-Large setting and the ELECTRA setting (Clark et al., 2019b) in Appendix B and C.

## 4.1 EXPERIMENTAL DESIGN

We use BERT-Base (110M parameters) architecture for all experiments. Specifically, BERT-Base is consist of 12 Transformer layers. For each layer, the hidden size is set to 768 and the number of attention head is set to 12. To compare with TUPE-A and TUPE-R, we set up two baselines correspondingly: BERT-A, which is the standard BERT-Base with absolute positional encoding (Devlin et al., 2018); BERT-R, which uses both absolute positional encoding and relative positional encoding (Raffel et al., 2019) (Eq. (5)).

Following Devlin et al. (2018), we use the English Wikipedia corpus and BookCorpus (Zhu et al., 2015) for pre-training. By concatenating these two datasets, we obtain a corpus with roughly 16GB in size. We set the vocabulary size (sub-word tokens) as 32,768. We use the GLUE (**G**eneral **L**anguage **U**nderstanding **E**valuation) dataset (Wang et al., 2018) as the downstream tasks to evaluate the performance of the pre-trained models. All codes are implemented based on *fairseq* (Ott et al., 2019) in *PyTorch* (Paszke et al., 2017). All models are run on 16 NVIDIA Tesla V100 GPUs with mixed-precision (Micikevicius et al., 2017).

## 4.2 OVERALL COMPARISON

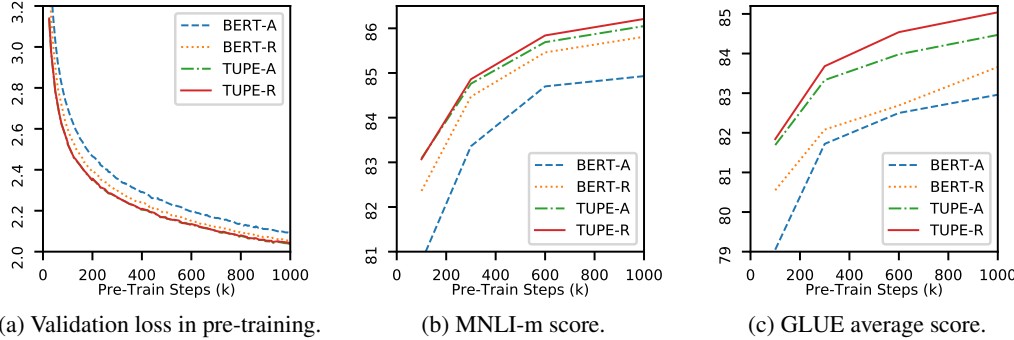

(a) Validation loss in pre-training.  (b) MNLI-m score.  (c) GLUE average score.

Figure 5: Both TUPE-A and TUPE-R converge much faster than the baselines, and achieve better performance in downstream tasks while using much fewer pre-training steps.

The overall comparison results are shown in Table 1. Firstly, it is easy to find that both TUPE-A and TUPE-R outperform baselines significantly. In particular, TUPE-R outperforms the best baseline BERT-R by 1.38 points in terms of GLUE average score and is consistently better on almost all tasks, especially on MNLI-m/mm, CoLA and MRPC. We can also see that TUPE-R outperforms TUPE-A by 0.57 points. As discussed in Sec.3.3, although using the absolute/relative positional encoding together seems to be redundant, they capture complement information to each other.

Besides the final performance, we also examine the efficiency of different methods. As shown in Figure 5a, TUPE-A (TUPE-R) achieves smaller validation loss than the baselines during pre-training. As shown in Table 1 and Figure 5c, TUPE-A (TUPE-R) can even achieve a better GLUE average score than the baselines while only using 30% pre-training steps. Similar improvements can be found in BERT-Large and ELECTRA settings in Table. 3 and Table. 4 (in Appendix C).

Since the correlations between words and positions are removed in TUPE, we can easily visualize the attention patterns over positions, without considering the variability of input sentences. In TUPE-A (see Figure 6), we find there are mainly five patterns (from 12 heads): (1) attending globally; (2) attending locally; (3) attending broadly; (4) attending to the previous positions; (5) attending to the next positions. Interestingly, the model can automatically extract these patterns from random initialization. As there are some attention patterns indeed have strong local dependencies, our proposed method to untie [CLS] is necessary. Similar patterns could be found in TUPE-R as well.

To summarize, the comparisons show the effectiveness and efficiency of the proposed TUPE. As the only difference between TUPE and baselines is the positional encoding, these results indicate TUPE can better utilize the positional information in sequence. In the following subsection, we will examine each modification in TUPE to check whether it is useful.

## 4.3 ABLATION STUDY

**Untie the [CLS] symbol from other positions.** To study the improvement brought by untying [CLS], we evaluate a positional encoding method which removes the reset function in Eq. 10. We call it TUPE-A$^{\text{tie-cls}}$ and train this model using the same configuration. We also list the performance of TUPE-A$^{\text{tie-cls}}$ in Table 1. From the table, we can see that TUPE-A works consistently better than TUPE-A$^{\text{tie-cls}}$, especially for low-resource tasks, such as CoLA and RTE.

**Untie the correlations between positions and words.** Firstly, from Table 1, it is easy to find that TUPE-A$^{\text{tie-cls}}$ outperforms BERT-A. Since the only difference between TUPE-A$^{\text{tie-cls}}$ and BERT-A is the way of dealing with the absolution positional encoding, we can get a conclusion that untying the correlations between positions and words helps the model training. To further investigate this, we design another encoding method, BERT-A$^d$, which is based on BERT-A and uses different projection matrices for words and positions. Formally, $\alpha_{ij} = \frac{1}{\sqrt{4d}} \left( (x_i^l W^{Q,l})(x_j^l W^{K,l})^T + (x_i^l W^{Q,l})(p_j U^K)^T + (p_i U^Q)(x_j^l W^{K,l})^T + (p_i U^Q)(p_j U^K)^T \right)$ in BERT-A$^d$. Therefore, we can check whether using different projection matrices (BERT-A vs. BRET-A$^d$) can improve the model and whether removing word-to-position and position-to-word correlations (BRET-A$^d$ vs. TUPE-A$^{\text{tie-cls}}$) hurts the final performance. From the summarized results in Table 1, we find that TUPE-A$^{\text{tie-cls}}$ is even slightly better (0.17) than BERT-A$^d$ and is more computationally efficient[4]. BERT-A is the worst one. These results indicate that using different projection matrices improves the model, and removing correlations between words and positions does not affect the performance.

**Summary.** From the above analysis, we find that untying [CLS] helps a great deal for the low-resource tasks, such as CoLA and RTE. Untying the positional correlation and word correlation helps high-resource tasks, like MNLI-m/-mm. By combining them, TUPE can consistently perform better on all GLUE tasks. We also have several failed attempts regarding modifying the positional encoding strategies, see Appendix D.

---

[4]BERT-A$^d$ is inefficient compared to TUPE as it needs to additionally compute middle two correlation terms in all layers.

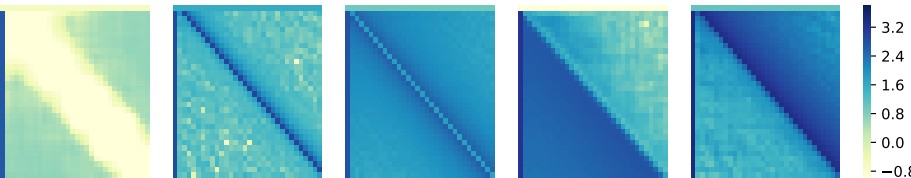

Figure 6: The visualization of learned positional correlations by TUPE-A.

## 5 RELATED WORK

As introduced in Sec.2, Shaw et al. (2018) was the first work to leverage relative positional encoding to Transformer. Most of the other works are based on Shaw et al. (2018). For example, Transformer-XL (Dai et al., 2019) re-parameterize the self-attention to integrate relative positional encoding directly. T5 (Raffel et al., 2019) simplified the vector representation of relative positions in Shaw et al. (2018) to a scalar. He et al. (2020) extended Shaw et al. (2018) by adding the position-to-word correlation for relative position. In (Kitaev & Klein, 2018), the authors show that separating positional and content information in the Transformer encoder can lead to an improved constituency parser. We mathematically show that such disentanglement also improves Transformer in general language pre-training. There are some other parallel works to enhance the absolute positional encoding in Transformer, but not directly related to our work. For example, Shiv & Quirk (2019) extended the sequence positional encoding to tree-based positional encoding in Transformer; Wang et al. (2019) extended positional encoding to complex-valued domain; Liu et al. (2020) modeled the positional encoding by dynamical systems.

## 6 CONCLUSION

We propose TUPE (Transformer with Untied Positional Encoding), which improves existing methods by two folds: untying the correlations between words and positions, and untying `[CLS]` from sequence positions. Specifically, we first remove the absolute positional encoding from the input of the Transformer and compute the positional correlation and word correlation separately with different projection matrices in the self-attention module. Then, we untie `[CLS]` by resetting the positional correlations related to `[CLS]`. Extensive experiments demonstrate that TUPE achieves much better performance on GLUE benchmark. Furthermore, with a better inductive bias over the positional information, TUPE can even outperform the baselines while only using 30% pre-training computational costs.

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

## A    PROOF OF PROPOSITION 1

*Proof.* We first construct a circulant matrix $\hat{B}$ of shape $2n \times 2n$ using $B$. Note that $B$ is a Toeplitz matrix consisting of $2n - 1$ values $\{b_{-(n-1)}, b_{-(n-2)}, \cdots, b_0, \cdots, b_{(n-2)}, b_{(n-1)}\}$. For ease of reference, we further define $b_{-n} = b_n = b_0$. Then $\hat{B}$ is constructed as

$$\hat{B}_{j,k} = \begin{cases} b_{k-j} & -n \leq k - j \leq n \\ b_{k-j-2n} & n < k - j < 2n \\ b_{k-j+2n} & -2n < k - j < -n \end{cases}. \tag{12}$$

To avoid any confusion, we use $k$ and $j$ in the subscription for positions and use $i$ as the imaginary unit. It is easy to check that $\hat{B}$ is a circulant matrix by showing that $\hat{B}_{j,k} = \hat{B}_{j+1,k+1}$ and $\hat{B}_{j,2n-1} = \hat{B}_{j+1,0}$ for any $0 \leq j, k < 2n - 1$. Using Theorem 7 in Gray (2006), the matrix $\hat{B}$ can be factorized into $\hat{B} = QDQ^*$. $Q$ is an $2n \times 2n$ Vandermonde matrix, where $Q_{k,j} = \frac{1}{2n}e^{i\pi(j+1)k/n}$ and $D$ is a $2n \times 2n$ diagonal matrix. Since the top-left $n \times n$ submatrix of $\hat{B}$ is $B$, we can rewrite $\hat{B} = \begin{bmatrix} B & C_1 \\ C_2 & C_3 \end{bmatrix}$. We also rewrite $Q = \begin{bmatrix} G \\ C_4 \end{bmatrix}$, where $G$ is defined in the theorem. Then we have $QDQ^* = \begin{bmatrix} G \\ C_4 \end{bmatrix} \begin{bmatrix} D_1 & 0 \\ 0 & D_2 \end{bmatrix} [G^* \quad C_4^*]$. Considering that $D$ is a diagonal matrix, we can obtain $B = GDG^*$ using block matrix multiplication. $\square$

## B    EXPERIMENTAL DETAILS

**Pre-training.**    Following BERT (Devlin et al., 2018), we use the English Wikipedia corpus and BookCorpus (Zhu et al., 2015) for pre-training. By concatenating these two datasets, we obtain a corpus with roughly 16GB in size. We follow a couple of consecutive pre-processing steps: segmenting documents into sentences by Spacy[5], normalizing, lower-casing, and tokenizing the texts by Moses decoder (Koehn et al., 2007), and finally, applying byte pair encoding (BPE) (Sennrich et al., 2015) with setting the vocabulary size as 32,768.

We found the data cleaning is quite important for language pre-training. Specially, we de-duplicate the documents, normalize the punctuations, concatenate the short sequences, replace the URL and other hyperlinks to special tokens, and filter the low-frequency tokens. Therefore, our re-implemented baselines, like BERT, and achieve higher GLUE scores than the original papers.

We use masked language modeling as the objective of pre-training. We remove the next sentence prediction task and use *FULL-SENTENCES* mode to pack sentences as suggested in RoBERTa (Liu et al., 2019). We train the models for $1000k$ steps where the batch size is 256 and the maximum sequence length is 512. The masked probability is set to 0.15, with replacing 80% of the masked positions by `[MASK]`, 10% by randomly sampled words, and keep the remaining 10% unchanged. We use Adam (Kingma & Ba, 2014) as the optimizer, and set the its hyperparameter $\epsilon$ to 1e-6 and $(\beta 1, \beta 2)$ to (0.9, 0.999). The peak learning rate is set to 1e-4 with a $10k$-step warm-up stage. After the warm-up stage, the learning rate decays linearly to zero. We set the dropout probability to 0.1, gradient clip norm to 1.0, and weight decay to 0.01. Besides the final checkpoint, we also save intermediate checkpoints ($\{100k, 300k, 600k\}$ steps) and fine-tune them on downstream tasks, to check the efficiency of different methods.

**Fine-tuning.**    We use the GLUE (**G**eneral **L**anguage **U**nderstanding **E**valuation) dataset (Wang et al., 2018) as the downstream tasks to evaluate the performance of the pre-trained models. Particularly, we use nine tasks in GLUE, including CoLA, RTE, MRPC, STS, SST, QNLI, QQP, and MNLI-m/mm. For the evaluation metrics, we report Matthews correlation for CoLA, Pearson correlation for STS-B, and accuracy for other tasks. We use the same optimizer (Adam) with the same hyperparameters as in pre-training. Following previous works, we search the learning rates during the fine-tuning for each downstream task. The setting details are listed in Table 2. For a fair comparison, we do not apply any tricks for fine-tuning. Each configuration will be run five times with different random seeds, and the *median* of these five results on the development set will be used as the performance of one configuration. We will ultimately report the best number over all configurations.

---

[5]https://spacy.io

Table 2: Hyperparameters for the pre-training and fine-tuning.

|  | Pre-training | Fine-tuning |
|---|---|---|
| **Max Steps** | $1M$ | - |
| **Max Epochs** | - | 5 or 10 [a] |
| **Learning Rate** | 1e-4 | {2e-5, 3e-5, 4e-5, 5e-5} |
| **Batch Size** | 256 | 32 |
| **Warm-up Ratio** | 0.01 | 0.06 |
| **Sequence Length** | 512 | 512 |
| **Learning Rate Decay** | Linear | Linear |
| **Adam $\epsilon$** | 1e-6 | 1e-6 |
| **Adam ($\beta_1$, $\beta_2$)** | (0.9, 0.999) | (0.9, 0.999) |
| **Clip Norm** | 1.0 | 1.0 |
| **Dropout** | 0.1 | 0.1 |
| **Weight Decay** | 0.01 | 0.01 |

[a] we use five for the top four high-resource tasks, MNLI-m/-mm, QQP, and QNLI, to save the fine-tuning costs. Ten is used for other tasks.

Table 3: GLUE scores on dev set. Different models are pre-trained in the BERT-Large setting (330M) with 16GB data. TUPE-Large$^{\text{mid}}$ is the intermediate $300k$-step checkpoint of TUPE-Large.

|  | Steps | MNLI-m/mm | QNLI | QQP | SST | CoLA | MRPC | RTE | STS | Avg. |
|---|---|---|---|---|---|---|---|---|---|---|
| BERT-Large | $1M$ | 88.21/88.18 | **93.56** | 91.66 | 94.08 | 58.42 | **90.46** | 77.63 | 90.15 | 85.82 |
| TUPE-Large | $1M$ | **88.22/88.21** | 93.55 | **91.69** | **94.98** | **67.46** | 90.06 | **81.66** | **90.67** | **87.39** |
| TUPE-Large$^{\text{mid}}$ | $300k$ | 86.92/86.80 | 92.61 | 91.48 | 93.97 | 63.88 | 89.26 | 75.82 | 89.34 | 85.56 |

**Normalization and rescaling.** Layer normalization (Ba et al., 2016; Xiong et al., 2020) is a key component in Transformer. In TUPE, we also apply layer normalization on $p_i$ whenever it is used. The $\frac{1}{\sqrt{d}}$ in Eq. (2) is used in the Transformer to rescale the dot product outputs into a standard range. In a similarly way, we use $\frac{1}{\sqrt{2d}}$ in the Eq. (7) to both terms to keep the scale after the summation. Furthermore, in order to directly obtain similar scales for every term, we parameterize $\theta_1$ and $\theta_2$ by using $\theta_1 = \frac{1}{\sqrt{2d}}(p_{\theta_1}U^Q)(p_{\theta_1}U^K)^T$ and $\theta_2 = \frac{1}{\sqrt{2d}}(p_{\theta_2}U^Q)(p_{\theta_2}U^K)^T$, where $p_{\theta_1}, p_{\theta_2} \in \mathbb{R}^d$ are learnable vectors.

## C  MORE RESULTS IN THE BERT-LARGE/ELECTRA-BASE SETTINGS

In the main body of the paper, we provide empirical studies on TUPE under the BERT-Base setting. Since TUPE only modifies the positional encoding, one expects it to improve all language pre-training methods and deeper models. To demonstrate this, we integrate TUPE-R (with relative position) in BERT-Large and ELECTRA-Base models, and conduct experiments for comparison. For the BERT-Large setting, we directly apply TUPE-R to the 24-layer Transformer model and obtain the TUPE-Large model. For ELECTRA-Base, we apply our positional encoding to both generator and discriminator and obtain the ELECTRA-TUPE model. We use the same data introduced previously and use the same hyper-parameters as in Liu et al. (2019); Clark et al. (2019b). The experimental results are listed in the Table. 3 and Table. 4 respectively. From the results, we can that TUPE is better than the corresponding baseline methods. These additional experiments further demonstrate that TUPE is a better positional encoding solution.

## D  FAILED ATTEMPTS

We tried to replace the parametric form of the positional correlation ($\frac{1}{\sqrt{2d}}(p_iU^Q)(p_jU^K)^T$) to the non-parametric form. However, empirically we found that the training of this setting converges much slower than the baselines. We also tried to parameterize relative position bias $b_{j-i}$ by $(r_{j-i}F^Q)(r_{j-i}F^K)^T$. But the improvement is just marginal.

Table 4: GLUE scores on dev set. Different models are pre-trained in the ELECTRA-Base setting (120M) with 16GB data. Electra-TUPEmid is the intermediate $600k$-step checkpoint of ELECTRA-TUPE.

| | Steps | MNLI-m/mm | QNLI | QQP | SST | CoLA | MRPC | RTE | STS | Avg. |
|---|---|---|---|---|---|---|---|---|---|---|
| ELECTRA | $1M$ | 86.42/86.14 | **92.19** | 91.45 | 92.97 | 65.92 | **89.74** | 73.56 | 89.99 | 85.38 |
| ELECTRA-TUPE | $1M$ | **86.98/87.01** | 92.03 | **91.75** | **93.97** | **66.51** | **89.74** | **76.53** | **90.13** | **86.07** |
| ELECTRA-TUPEmid | $600k$ | 86.72/86.74 | 91.84 | 91.72 | 93.30 | 66.42 | 89.34 | 75.64 | 89.67 | 85.71 |

