# OpenReview forum: "Rethinking Positional Encoding in Language Pre-training"
_ICLR.cc/2021/Conference — ICLR 2021 Poster_

### Official Review · AnonReviewer2 · 2020-10-28

**Rating:** 6
**Confidence:** 4

**Review:**

This paper studies the positional encoding in BERT. It argues against the word—position correlations that are implicitly imposed by BERT’s treatment of positional encodings. The paper proposes to decouple the “content attention” (token to token) and “contentless attention” (position to position), and remove the so called heterogeneous interactions between tokens and positions. Further, it points out that it is problematic to treat the special [CLS] symbol as a token in calculating its positional encoding: per standard practice, [CLS]’s representation is used as a summary of the full sequence, and treating it as a token implicitly biases the sentence representation towards those tokens close [CLS]. To resolve this issue, the paper “hides” the position information of [CLS]. The proposed approach is built on top of BERT and evaluated on  the GLUE benchmark. Experimental results show that it outperforms the baselines.

Overall I find this paper interesting and clearly-presented. The experiments are solid and well-executed too. My main concern is that the intuition that “a word does not have strong correlations to its position,” which is one of the key motivations, can be better elaborated (see details below).

Pro:
- The proposed solutions to the issues pointed out by the paper are clean and reasonable.
- Untying [CLS]’s positional encoding is insightful.
- Strong performance.
- Writing is very clear.

Con:
- The key motivation that a word has little to do with where it appears is debatable.
- It’s hard to figure out whether or not the improvement could come from using absolute positional encodings at every layer.

Details:
- Upfront the paper argues that “without specific context there is little evidence that a word has a strong correlation to where it appears in the sentence,” and cites Köhler et al., (2008) for support. First of all, I find this claim, to the best of my knowledge, counterintuitive and misleading. I am not able to verify this claim from the cited work: it seems to be a book about linguistics and is written in German, which I do not have any expertise. Since this is the key motivation of this paper, I encourage the authors to elaborate it: (1) Please explain what it means by “without specific context.” Clearly in “contextualized” pretraining every word appears in context. (2) Clarify what “correlation” refers to from a linguistic perspective. Footnote 1 indicates that it refers to dot product between vectors, which does not seem to fit in the linguistic argument the paper is trying to make. (3) Verify that this phenomenon, if true at all, applies to all languages or specify the languages it applies. (4) If at all possible, please additionally cite a supporting work written in English so that it is more accessible to a broader audience, especially to those without German expertise like myself.
- The paper empirically supports the above claim using a visualization experiment in Figure 5. However, I find it hard to interpret without knowing the experimental details. Does this apply to all layers? Is it consistent across all attention heads? Do the authors think this is due to the learning biases of BERT and transformers, or it is an inherent property of the data and the English language?
- Again about this claim: I don’t think it is well-supported in the experiments either: comparing BERT-A^d against TUPE-A^{tie-cls}, less than 0.15 average improvement can be attributed to removing word-position correlation, and the performance is mixed across the datasets.
- Adding onto the above three points: this claim serves as a motivation instead of a key research question. So I don’t think the paper’s contribution will be weaker if it tones down this claim or does not talk about it at all.
- BERT uses the positional encoding at the first layer and propagate it all the way up with the residual connections. In contrast, some works use relative positional encodings at every layer. Equation 7 suggests that TUPE uses absolute positional encodings at every layer. Can the authors comment on how this compares to (1) a TUPE model that uses positional encoding only at the first layer (so that it is more comparable to BERT) and (b) a BERT model that uses absolute positional encodings at every layer?

---

> ### Author Response · Authors · 2020-11-17
> **Response to AnonReviewer2**
>
> Thanks very much for supporting our work and kind suggestions! We answer each of your questions below.
>
> 1. Clarification of "without specific context there is little evidence that a word has a strong correlation to where it appears in the sentence"
>
> Thanks for the question. We made more discussions in an early version of the paper but removed them due to the paper length limit. We provide more explanation below.
>
> In BERT and recently developed advanced methods such as RoBERTa, sentences are patched in a random way, making the word and position have weaker correlations. In detail, during pre-training, each input contains multiple sentences that are not required to be "contiguous" strictly (See the negative sampling of sentences for the next sentence prediction task in BERT and the full-sentence mode in RoBERTa). Such a strategy is shown to be more efficient and effective. This data pre-processing strategy is also applied to multi-lingual data corpus and works well for multiple languages.
>
> It can be easily seen that due to the random process of batching, the same sentence can appear at either the early or late part of the input sequence in different epochs, so do the words.
>
> We will make a better statement in the rebuttal version of the submission.
>
> 2. Regarding the visualization details and interpretation
>
> The visualization (figure 5) is provided for the four correlations in Equation (6). See the caption of figure 5. That being said, we visualize the four terms in the first attention layer in a well-trained BERT model. The phenomenon is similar across all attention heads.
>
> For deeper layers, it is hard to make such a visualization correctly as each layer takes multiple non-linear transforms to the input (word embedding+positional embedding). It is impossible to well separate the semantic signals from the positional signals and compare different types of correlations.
>
> We guess the pattern is likely to be learned from the language data as all the Transformer parameters are randomly initialized at the beginning without any specific bias.
>
> 3. Regarding the comparison between BERT-A^d against TUPE-A^{tie-cls}
>
> Thanks for the careful review. We agree that improvement (0.17) seems not significant. But this experimental result shows that the two terms in the middle *might be useless* as additionally using them in the attention (BERT-A^d) does not affect the final performance much (compared to TUPE-A^{tie-cls}). Furthermore, we also want to highlight that BERT-A^d is computationally inefficient as it has to compute the two correlation terms in the middle in every layer, while TUPE doesn't need these.
>
> We will make a better statement in the rebuttal version of the submission.
>
>
> 4. Additional experiments on (a) a TUPE which uses positional encoding only at the first layer  and (b) a BERT model that uses absolute positional encodings at every layer
>
> We empirically find that our proposed absolute positional encoding should be used in each layer, like relative positional encoding. Only using it in the first layer and removing it in the upper layer (your experiment (a)) hurts the performance. We think this result is reasonable as our proposed absolute positional encoding acts like a bias term directly in the attention model. Removing this proper bias in upper layers makes the model worse.
>
> BERT-A^d is already close to your experiment (b) as we explicitly use absolute positional encodings at every layer (but with different projection matrices for word and position). Before this submission, we tried the simple per-layer absolute positional encodings in BERT, and it has slight improvements (worse than BERT-A^d).

---

> > ### Comment · AnonReviewer2 · 2020-11-17
> > **Thanks for the detailed response!**
> >
> > The response addressed my concerns. I have read other reviews, and would like to keep my initial score. It would be great if the authors can be very careful about the claims on word-position correlation, and be more explicit that this is a consequence of how we train contextualized embeddings, rather than a universal linguistic phenomenon. Otherwise readers might draw misleading conclusions from the paper.

---

> > > ### Author Response · Authors · 2020-11-18
> > > **Thanks for the quick feedback!**
> > >
> > > We will revise our paper accordingly and update it before the end of the rebuttal period.

---

### Official Review · AnonReviewer1 · 2020-10-29
**Nice analysis and improvement on Transformer's position encoding**

**Rating:** 7
**Confidence:** 4

**Review:**

Summary: The paper delves into the nature of positional encoding in Transformer and variants (especially BERT). The paper points out that, attention weight computation obtained from the addition of the word embedding and the position embedding in the first layer can be expanded into four terms, namely word-word, position-position, and two word-position (in both directions). The paper points out that word-position relationship is largely meaningless, which is also demonstrated via visualizing the values in pertained BERT. Then the paper proposes to remove these two terms when computing the attention. Secondly, the paper mentions that special tokens in BERT (e.g. [CLS]) should be treated in a different way than typical words. To do so, it proposes to replace the attention logic with a learnable parameter if the attention involves one or two special tokens. Both methods combined show a significant advantage on multiple GLUE tasks when applied to BERT.

Strengths:
- The observations on the issues of the PE in BERT (Transformer) are interesting and convincing.
- The proposed method based on the observations result in significant experimental benefit in multiple classification datasets in GLUE.

Weaknesses:
- I am wondering if all experiments in Table 1 were done with exactly the same setup including random seed. This is because BERT was known for being undertrained and some slight changes with respect to the original BERT could result in non-trivial gains.
- It would be good to see how the proposed method works in other more recent pertained models including RoBERTa.

Overall: I think the paper nicely touches on positional encoding, an important component of Transformer-based architectures, and the experimental results are encouraging.

---

> ### Author Response · Authors · 2020-11-17
> **Response to AnonReviewer1**
>
> Thanks very much for supporting our work!
>
> Regarding your questions, first, we did use the same seeds for all the models during pre-training and fine-tuning. During fine-tuning, we use five seeds {1,2,3,4,5} for each pre-trained model and report the average performance.
>
> Second, in the appendix section, we have provided more experimental results using TUPE to the recent ELECTRA model (See Table 4) and BERT-large (See Table 3). From the results, we can see that our method consistently improves the backbone models, suggesting that TUPE can be combined with different pre-training methods broadly. Note that in our baseline models, we have already applied the suggestions in the #RoBERTa# paper for pre-training (See Appendix B, pre-training paragraph). We just called the baselines generally as #BERT# in the paper.

---

### Official Review · AnonReviewer3 · 2020-10-30
**An improvement to positional encoding for contextual language models**

**Rating:** 7
**Confidence:** 4

**Review:**

This paper introduces an approach for the positional encoding of input tokens into pretrained contextual embedding models. Specifically, they compute the positional correlation and the embedding separately, then add them together. In addition, they treat the [CLS] symbol separately from the other positions, which is a long-overdue improvement.

This paper's motivation is clear, they derive intuitive improvements for the positional encoding of inputs to BERT. Writing out the absolute and relative positional encodings was useful -- they motivate the untying by showing that the absolute positional encoding is actually composed of four terms in equation 6, two of which are not encoding useful information. They then introduce in equation 9 their modification to the positional encoding of the CLS symbol. Again, this is fairly straightforward, and likely leaves room for improvement but is a reasonable first step.

Their experiments support their hypotheses that this builds a better representation than the standard positional encoding schemes. They present results on the GLUE dev set, and while for some of the smaller datasets in GLUE results can be sometimes unstable (even between runs changing only the random seed), the performance improvements are so consistent that there is clear evidence in favor of the hypothesis that this change is an improvement.

In general, I don't see any clear faults, and I expect this work will be built on quickly or fairly widely adopted. It's novel, clear, and as far as I'm aware original, and even though it's a relatively simple idea, it's well supported by the experimental evidence. It's possible adding other LMs to the experiments would add to the strength of the argument, but to me this is enough.

Edit: After reviewing the author response, I leave my score unchanged. I believe this paper should be accepted.

---

> ### Author Response · Authors · 2020-11-17
> **Response to AnonReviewer3**
>
> Thanks very much for supporting our work!
>
> In the appendix section, we have provided more experimental results using TUPE to ELECTRA and BERT-large (See Table 4). From the results, we can see that our method consistently improves the backbone models, suggesting that TUPE can be combined with different pre-training methods broadly.

---

### Official Review · AnonReviewer4 · 2020-11-03
**Advancement to Transformer Architecture**

**Rating:** 7
**Confidence:** 4

**Review:**

The paper takes a stab at fixing the correlation problem between the positional encoding and word embedding when they are added together in a transformer. The paper fixes this by untying the position encoding from the word embedding and passing them through separate routes of projection key and value matrices before merging them. The paper also tackles the problem of untying [CLS] token from rest of the position tokens due to its significance in downstream tasks. Authors claim that [CLS] token if not properly tied is susceptible to only gathering information from local starting words in the sentence. The authors fix this by resetting the positional correlations related to CLS from others and to others from [CLS]. The authors provide nice theoretical intuitions behind their claims which I find justified.

The proposed method is called TUPE short for Transformers with Untied Positional Encoding. TUPE outperforms the BERT baseline easily on GLUE tasks and shows empirically that this was quite necessary and semantically word embeddings and position embedding were indeed not similar. I recommend this paper for acceptance as this is an important advancement in Transformer architecture which is becoming a de-facto standard in NLP and elsewhere.

Questions to authors:
- It would be great if qualitative analysis similar to Figure 4. Can be shown for original correlation in positional embeddings and word-embeddings to qualitatively support the intuitions. Similarly, for the original [CLS] token.

- Original BERT model was evaluated on SQuAD and SWAG as well. Do these claims hold there as well?

Edit after rebuttal: I have read the author response and I am thankful to authors for answering my questions. I keep my rating as it is and believe that the paper should be accepted. I hope this will be widely adopted in NLP community.

---

> ### Author Response · Authors · 2020-11-17
> **Response to AnonReviewer4**
>
> Thanks very much for supporting our work!
>
> Figure 5 in the appendix is exactly the qualitative analysis of the original model that you expect to have. From the second and third sub-figures, we can see that the correlations between a word and a position are not strong since the values in the figures look uniform. Furthermore, in all the figures, the values in the first row/column are the correlations related to the [CLS] token. It can be seen that the [CLS] token embedding has special patterns in the word-word correlation matrix (sub-figure 1), but the [CLS] positional embedding does not have such patterns (sub-figure 4). Both statistics support our intuition.
>
> Our implementation is based on fairseq, which doesn't officially support SQuAD/SWAG. We have been working on implementing SQuAD on fairseq on our own and will update the results once they are ready. We also tried TUPE on some question-answering matching tasks (binary ranking) and observed consistent improvements.

---

### Author Response · Authors · 2020-11-23
**Summary of paper revision**

We thank AC for handling this paper and thank all reviewers for the useful suggestions. We updated a new version according to the review comments and here are the main changes.

* We increase the paper length to nine pages by moving some relevant figures from the appendix to the main body.

* We rewrite the discussion about the position-word correlations and the empirical comparison between BERT^Ad and TUPE-tie-cls.

Thanks

Paper772 Authors

---

### Decision · Program_Chairs · 2021-01-07
**Final Decision**

**Decision:**

Accept (Poster)

**Comment:**

This paper revisits the design of positional embedding in the pre-trained language models, and propose a new approach to handling the positional encoding.

Overall, the paper is well-motivated. The authors have addressed most comments based on the review. The method proposed in the paper is simple and effective. Experiments are comprehensive and demonstrate the effectiveness of the proposed approaches.